# Variation in Phenolic Profile, Antioxidant, and Anti-Inflammatory Activities of *Salvadora oleoides* Decene. and *Salvadora persica* L. Fruits and Aerial Part Extracts

**DOI:** 10.3390/life12091446

**Published:** 2022-09-18

**Authors:** Arifa Khanam, Ashfaq Ahmad, Neelam Iftikhar, Qasim Ali, Tabinda Fatima, Farhan Khashim Alswailmi, Abdullah Ijaz Hussain, Sulaiman Mohammed Abdullah Alnasser, Jamshaid Akhtar

**Affiliations:** 1Department of Chemistry, Government College University Faisalabad, Faisalabad 38000, Pakistan; 2Department of Pharmacy practice, College of Pharmacy, University of Hafr Al Batin, Hafr Al Batin 31991, Saudi Arabia; 3Department of Botany, Government College University Faisalabad, Faisalabad 38000, Pakistan; 4Department of Pharmaceutical Chemistry, College of Pharmacy, University of Hafr Al Batin, Hafr Al Batin 31991, Saudi Arabia; 5Department of Pharmacology and Toxicology, Unaizah College of Pharmacy, Qassim University, Buraidah 51452, Saudi Arabia; 6Department of Internal Medicine, Allama Lqbal Medical College, Lahore 54700, Pakistan

**Keywords:** phenolic acid, flavonoids, DPPH radical scavenging activity, anti-inflammatory activity, maximum possible analgesia

## Abstract

(1) Background: The objective of this study was to investigate the potential of *Salvadora oleoides* (*S. oleoides*) and *Salvadora persica* (*S. persica*) polyphenols as antioxidant and anti-inflammatory agents. (2) Methods: Aerial parts and fruits of *S. oleoides* and *S. persica* were collected from the periphery of District Bhakkar, Punjab, Pakistan. Methanol extracts were prepared using the Soxhlet extraction technique. Extract yield varied from 8.15 to 19.6 g/100 g dry plant material. RP-HPLC revealed the detection of thirteen phenolic aids and five flavonoids. Gallic acid, hydroxy benzoic acid, chlorogenic acid, and cinamic acid were the major phenolic acids, whereas catechin, rutin, and myricetin were the flavonoids detected. (3) Results: Maximum total phenolic contents (TPCs) (22.2 mg/g of dry plant material) and total flavonoid contents (TFCs) (6.17 mg/g of dry plant material) were found in the fruit extract of *S. persica*, and the minimum TPC (11.9 mg/g) and TFC (1.72 mg/g) were found in the aerial part of *S. oleoides.* The fruit extract of *S. persica* showed the highest DPPH radical scavenging activity. In vivo anti-inflammatory activity of all the extracts was performed on albumin-induced rat paw edema that was comparable with the standard indomethacin; *S. persica* fruit extract showed remarkable anti-inflammatory activity. Analgesic activity of aerial part and fruit extracts of *S. oleoides* and *S. persica* was investigated using a mouse model, and the results showed that maximum possible analgesia of fruit extracts of *S. persica* was 53.44%, which is better than the PC group (52.98%). (4) Conclusions: The variations in the antioxidant, anti-inflammatory, and analgesic activities of methanolic extracts of *S. oleoides* and *S. persica* were found to be significant, and they have therapeutic potential as antioxidant, analgesic, and anti-inflammatory agents.

## 1. Introduction

Reactive oxygen species produced during different metabolic activities cause oxidative damages that are responsible for many diseases including, Parkinson’s and Alzheimer’s diseases, tumor formation, heart diseases, nervous disorders, pulmonary disorders, rheumatic premature aging, and inflammatory diseases [1,2]. Inflammation has multiple actions in the process of growth and differentiation, as well as in lymphoid and non-lymphoid cells, and regulates their production during injury and infections [3]. Chronic inflammation can cause the destruction of tissues and cells [4]. Antioxidants, especially natural compounds, can combat against oxidative stress and inflammation that provide relief against degenerative diseases [5,6,7].

The Asia-Pacific region is rich in plant genetic resources, including less known food plants and underutilized species that serve as a means of survival during times of famine, shocks, drought, and risks. They also contribute immensely to family food security, which can supplement nutritional requirements due to their better nutritional value [8]. Furthermore, these medicinal plants are potential sources of natural products that are generally recognized as safe (GRAS) [7]. The standardized extracts of these plants/isolated compounds, especially the polyphenols, provide unlimited opportunities for new drug discoveries [5,6,9]. Huge proportions of the global population are now more dependent on natural and alternative traditional medicines as primary health care [9,10].

The family Salvadoraceae has some underutilized species that are little known, despite their wide distribution in arid areas around the world [11]. In Pakistan, the genera Salvadora has two underutilized species, i.e., *Salvadora persica (S. persica)* and *Salvadora oleoides (S. oleoides)* [12]. *S. persica* (arak, jhak, pilu) is a small tree or shrub known by the name miswak [13]. Pharmacological studies indicated that the *S. persica* plant possesses alexiteric, analgesic, anti-inflammatory, anti-microbial, anti-plaque, astringent, diuretic aphrodisiac anti-pyretic, and bitter stomachic activities [14]. In folk literature, it is used by various populations worldwide, especially by Muslims, and it cited in the Holy Quran [15,16]. *S. oleoides* (Jall, Mitha Jall, Peelu, Pilu, Khabbar) is an oil-yielding medicinal and multipurpose tree, which is reported to possess anti hypoglycemic, hypolipidemic, analgesic, anti-inflammatory and antimicrobial activities [17]. *S. oleoides* plant extract contains terpenoids, phenolic compounds, alkaloid, glycosides, and flavonoids, which are frequently used against many microbial activities [18]. 

Currently, due to the many side effects of synthetic medicines, the public is now again shifting towards natural products due to their GRAS status and negligible side effects. Hence, there is contemporary need to explore more underutilized species having strong traditional use against specific diseases [19]. To the best of our knowledge, no previous reports have been presented in the literature on the comparative evaluation of antioxidant and anti-inflammatory activities of aerial parts and fruits of *S. oleoides* and *S. persica*. Therefore, this study was planned to investigate the antioxidant and anti-inflammatory activities of polyphenol-rich fractions of *S. oleoides* and *S. persica* aerial parts and fruits. Moreover, simultaneous quantifications of phenolic acids and flavonoids were performed using reverse-phase high-performance liquid chromatography (Rp-HPLC). The in vivo anti-inflammatory and analgesic activities were assessed using albumin-induced inflammation in rat paw and the measurement of maximum possible analgesia in mice models.

## 2. Materials and Methods

### 2.1. Collection and Identification of Plant Materials

Fruits and aerial parts (10 kG) of *S. oleoides* Decene. and *S. persica* L. were collected from the periphery of District Bhakkar during summer in May and June, 2018. The samples were further authenticated by the Taxonomist, Department of Botany, Government College University Faisalabad, Pakistan. The samples were transferred to the Natural Product and Synthetic Chemistry Lab, Government College University, for further investigations.

### 2.2. Reagents and Chemicals

All the chemicals and reagents, including Folin–Ciocalteu phenolic reagent, gallic acid, 2, 2-diphenyl-1 picrylhydrazyl (DPPH), catechin, gallic acid, phenolic acid and flavonoid standards, ascorbic acid, and linoleic acid (75%), were purchased from Sigma-Aldrich Co., (St Louis, MO, USA). All the solvents, butylated hydroxyl toluene (BHT), hydrochloric acid, sodium carbonate, sodium hydroxide, aluminum chloride, and sodium nitrite, were purchased from Merck (Darmstadt, Germany). All chemicals used, including the solvents, were of analytical grade and used without further purification.

### 2.3. Preparation of Plant Extracts

Plant materials were dried in the shade and then powdered (80 mesh) using an electric grinder (BL 999SP, LG, Frankfurt, Germany). Extractions were carried out using the 500 mL Soxhlet extractor with absolute methanol, as reported previously [20]. Briefly, 50 g of powdered material was taken in a thimble and extracted with 300 mL of absolute methanol for 8 h on Soxhlet extraction apparatus. The extracts were then concentrated using a rotary evaporator (Eyela, SB-651, Rikakikai Co., Ltd., Tokyo, Japan) under reduced pressure and stored in a refrigerator at 4 °C for further studies. The percentage yield of each extract was determined using the formula given below.
$$\mathrm{Yield}\;\left(\frac{g}{100\;g}\right)=\frac{Weight\;of\;dry\;extract}{Weight\;of\;dry\;plant\;material}\times 100$$

### 2.4. Qualitative and Quantitative Analysis of Phenolic Acids and Flavonoids Simultaneously

Standard stock solutions of all the phenolic acid and flavonoids available were prepared fresh by dissolving the compounds in methanol (10 mg/mL). Working solutions (0.2–1.0 mg/mL) were prepared and standard curves were constructed by plotting concentrations against peak areas. The extracts were prepared as reported previously and filtered through a 0.45 µm non-pyrogenic filter (Minisart, Satorius Stedim Biotech GmbH, Goettingen, Germany) prior to injection [7].

The HPLC analysis was performed with the Perkin Elmer system (Perkin Elmer, Japan) equipped with gradient model binary pump systems, and a UV/Visible detector. The injection mode was manual, and the degasser (DGU-20A5) system was intact. The column oven was installed and equipped with hypersil GOLD C18 column (250 × 4.6 mm internal diameter, 5 mm particle size) (Thermo Fischer Scientific Inc., Waltham, MA, USA) supported with a guard column and a non-linear gradient consisting of solvent A (acetonitrile: methanol, 70:30) and solvent B (water with 0.5% glacial acetic acid). The quantification was based on an external standard method, whereas the analytes were identified by matching the retention times and spiking the samples with the standard.

### 2.5. In Vitro Antioxidant Analysis

#### 2.5.1. Total Phenolic and Total Flavonoid Contents

Total phenolic contents (TPCs) and total flavonoid contents (TFCs) of the prepared extracts were measured according to the Folin–Ciocalteu phenol reagent and aluminum chloride colorimetric assays, respectively, as reported by Hussain et al. [7]; several different dilutions (10–80 ppm) of gallic acid were prepared to create a calibration curve (Y = 0.0265X − 0.1834). Similarly, different dilutions of catechin (10–160 ppm) were prepared and the calibration curve was derived (Y = 0.0063X − 0.023). The TPC and TFC of the extracts were calculated from respective curves and reported as mg per gram of dry matter, expressed as gallic acid equivalent (GAE) and catechin equivalent (CE), respectively.

#### 2.5.2. DPPH Radical Scavenging Assay

DPPH (2, 2-Diphenyl-1-picrylhydrazyl) radical scavenging activity was assayed by the method reported previously [7]. Briefly, 10 µg/mL concentrations of extracts and BHT were mixed with 2 mL of 90 μM DPPH. The solution was incubated at room temperature for half an hour. The absorbance was read at 517 nm, and scavenging in terms of the percentage was calculated as follows:$$\;\mathrm{Scavenging}\left(\%\right)=\frac{Absorbance\;of\;DPPH\;solution-Absorbance\;of\;sapmle\;solution}{Absorbance\;of\;DPPH\;solution}\times 100$$

### 2.6. In Vivo Evaluation of Anti-Inflammatory and Analgesic Activities

For in vivo experiments, all procedures were conducted in conformity with the international guidelines on the ethical use of animals, and the study was approved by Institutional Ethical Committee for Animal Care at Government College University Faisalabad (Study No 19680/IRB No 680).

#### 2.6.1. Animals

Adult male Wistar rats (weighing approximately 130–160 g) and mice weighing approximately 30–40 g were collected from the animal house of the Department of Physiology, Government College University, Faisalabad. Before the start of experiment, the animals were adapted for a week at the standard conditions (26 °C ± 2 °C temperature; 40–60% ambient humidity). Rats were housed in elevated wire cages with free access to food and water. The intake of food was measured on a daily basis for each animal.

#### 2.6.2. Anti-Inflammatory Activity

To examine the anti-inflammatory activity of *S. oleoides* and *S. persica* aerial parts and fruit extracts in rats, thirty-six albino male Westar rats were randomly divided into the following six groups, all having six rats in each group (n = 6).
NC GROUP: Normal control group, which received no treatment.PC GROUP: Positive control group, which were supplemented with a dose of 10 mg/kg body weight (bw) of indomethacin served as a standard drug.G1 GROUP: Treatment group-1, which were supplemented with 250 mg/kg bw extract of *S. oleoides* aerial parts.G2 GROUP: Treatment group-2, which were supplemented with 250 mg/kg bw extract of *S. persica* aerial parts.G3 GROUP: Treatment group-3, which were supplemented with 250 mg/kg bw extract of *S. oleoides* fruits.G4 GROUP: Treatment group-4, which were supplemented with 250 mg/kg bw extract of *S. persica* fruits.

All the rat groups were provided water ad libitum and normal feed (approx. 20 g/rat/day). For measuring the anti-inflammatory activity, extract doses of *S. oleoides* and *S. persica* were administrated orally through gavage feeding tube (16–18 cm) for 7 days, while positive controls were orally administered indomethacin 10 mg/kg bw for 7 days before the induction of inflammation [21]. After 7 days of treatment, inflammation was induced in the right paw by injecting the 0.1 mL egg albumin in each treatment and PC group. The inflammation was measured for 4 h by screw gauge. The percentage Edema Inhibition was calculated using the NC group as a standard.

#### 2.6.3. Analgesic Activity

To evaluate the analgesic activity, the mice were randomly divided into the following six groups, all having five rats in each group (n = 5).
NC GROUP: Normal Control, which received no treatment.PC GROUP: Positive Control, which were supplemented with a dose of 10 mg/kg of Diclofenac sodium served as standard.G1 GROUP: Treatment group-1, which were supplemented with 250 mg/kg bw extract of *S. oleoides* aerial partsG2 GROUP: Treatment group-2, which were supplemented with 250 mg/kg bw extract of *S. persica* aerial partsG3 GROUP: Treatment group-3, which were supplemented with 250 mg/kg bw extract of *S. oleoides* fruitsG4 GROUP: Treatment group-4, which were supplemented with 250 mg/kg bw extract of *S. persica* fruits.

Analgesic activity was performed as described by Hossain et al. [22]. Mice were fasted for 12 h with adequate clean water. Mice were placed on hot plates and the temperature was maintained at 55 ± 1 °C. Latency period, or the pain reaction time determined with a stopwatch, was recorded, which represented the time taken for the mice to react to the pain stimulus. The time in seconds was measured with a stopwatch by observing discomfort reactions, such as licking paws or jumping. The first reading was taken just before the administration of the drug and later at 0, 60, 120, 180 and 240 min before and after the drug administration. The cutoff time was fixed for 12 s to prevent any type of injury. This served as the control pain reaction time. The maximum possible analgesia (MPA) was calculated as follows:$$\mathrm{MPA}\;\left(\%\right)=\frac{Reaction\;time\;for\;treatment-Reaction\;time\;for\;saline}{12-Reaction\;time\;for\;saline}\times 100$$

### 2.7. Statistical Analysis

All the experiments were performed in three replicates and data are reported as the mean ± standard deviation (SD). Statistical analysis was performed by means of the statistical package, STATISTICA (Stat Sift Inc., Tulsa, OK, USA). Data from different tests were analyzed using one-way analysis of variance (ANOVA), followed by Bonferroni/Dunnett (all mean) post hoc tests; the differences between the means were considered statistically significant at probability value *p* ≤ 0.05.

## 3. Results

### 3.1. Yield of Extracts

The extract yields (g/100 g) from *S. oleoides* and *S. persica* aerial parts and fruits are given in Table 1. Extract yields varied from 8.15 to 19.60 g/100 g of dry plant material. The maximum extract yield (19.60 g/100 g) was obtained from the aerial part of *S. persica*, whereas the minimum extract yield (8.15 g/100 g) was found from *S. oleoides* aerial parts. The results showed a significant (*p* ≤ 0.05) difference in the yields among different extracts. Saleem et al. [23] reported a 12% extract yield from the aerial parts of *S. oleoides*. Variation in the extract yield might be due to differences in extractable components and months.

### 3.2. HPLC Results

Thirteen phenolic acids (gallic, hydroxy benzoic, chlorogenic, caffeic, syringic, vanillic, *p*-coumeric, salicylic, sinapic, ferulic, ellagic, cinamic, and benzoic acids) and five flavonoids (catechin, rutin, myricetin, quercetin, and kaempferol) were identified and quantified using Rp-HPLC from different extracts of *S. oleoides* and *S. persica*, and the data are presented in Table 2. Chlorogenic acid was the major phenolic acid from the aerial parts and fruit extracts of *S. oleoides* and *S. persica*, followed by gallic acid and hydroxyl benzoic acid. *S. oleoides* and *S. persica* fruit extracts exhibited the maximum concentrations of chlorogenic acids, i.e., 1786.0 and 1473.0 mg/100 g of extract, respectively. Gallic acid was abundantly found in the *S. oleoides* aerial part extract (1265.0 mg/100 g), followed by the *S. persica* fruit extracts (942.4 mg/100 g). *S. persica* fruit extracts also contained 942.4 mg/100 g hydroxyl benzoic acid, whereas *S. oleoides* fruit extract contained 727.0 mg/100 g hydroxyl benzoic acid. Cinamic acid was found abundantly in the aerial part extracts of *S. oleoides* (390.6 mg/100 g) and *S. persica* (471.6 mg/100 g) only (Figure 1). Catechin and myricetin were the major flavonoids detected in all extracts of *S. oleoides* and *S. persica*. *Salvadora persica* aerial part extracts were rich in myricetin, rutin, and catechin, and the contents of these were 410.5, 254.3, and 578.5 mg/100 g, respectively. *S. persica* fruit extracts contained 331.8, 275.1, and 117.2 mg/100 g myricetin, rutin, and catechin, respectively. Catechin was abundant in the aerial part extracts of *S. oleoides* (131.5 mg/100 g) and *S. persica* (578.5 mg/100 g) as compared with the fruit extracts.

Statistical analysis showed that the concentrations of various phenolic acids and the flavonoids were significantly (*p* ≤ 0.05) different among different extracts.

The phenolic and flavonoid contents were positively associated with the antioxidant activity. Both gallic acid and chlorogenic acid were the potential natural antioxidant compounds. Very few reports are available on the phenolic profile of the investigated extracts. Noumi et al. [24] also reported the separation and identification of caffeic acid, rutin trihydrate, trans-cinnamic, and gallic acids in the stems of *S. persica* using RP-HPLC.

### 3.3. Evaluation of In Vitro Antioxidant Activity

#### Estimation of TPC and TFC

TPC and TFC results, expressed as mg/g of dry plant material, are presented in Table 1. TPC was in the range of 11.9 to 22.2 mg TPC/g of dry plant material, measured as the GAE. The maximum TPC (22.2 mg/g of dry plant material, as the GAE) was found in fruit extracts of *S. persica*, and the minimum TPC (11.9 mg/g, as the GAE) was found in the aerial part of *S. oleoides*, followed by the aerial part extracts of *S. persica* (16.2 mg/g, as the GAE) and the fruit extracts of *S. oleoides* (19.2 mg/g, as the GAE). The TFCs in aerial parts and fruit extracts were in the range of 1.63 to 6.17 mg TFC/g of dry plant material, measured as the CE. The maximum amount of TFC (6.17 mg TFC/g of dry plant material as the CE) was exhibited by fruit extracts of *S. persica*, and the minimum TFC (1.72 mg TFC/g of dry plant material as the CE) was exhibited by aerial part extracts of *S. oleoides*. The flavonoid contents in fruit extracts of *S. oleoides* were found to be 5.54 mg TFC/g of dry plant material and 2.44 mg TFC/g of dry plant materials in aerial part extracts of *S. persica*. Significant differences (*p* ≤ 0.05) were observed in the TPC and TFC of the different extracts of *S. oleoides* and *S. persica*.

Phenolic compounds have good relationship and largely contribute to the antioxidant activity. The polar solvent, i.e., methanol, which was used has the capacity to extract more phenolics as compared with other solvents. Saleem et al. [23] reported total phenolic (0.4 mg QE/g) and total flavonoid contents (0.21 mg QE/g) of *S. oleoides* aerial parts in methanol extract which were lower than our results. Kumari et al. [25] reported the total estimated amount of phenolics in the fruit methanol extract of *S. persica* to be 120.38 mg/100 g DW, and flavonoids were estimated to be 77.59 mg/100 g DW; however, there have been no reported results on the TPC and TFC of *S. oleoides* fruit. Kaneria et al. [19] reported that *S. persica* showed higher total phenol contents as compared with *S. oleoides*. The total phenolic contents of *S. oleoides* in methanol extracts of leaves were 253.10 mg/g, whereas the total flavonoid contents were 43.65 mg/g. The total phenolic content of *S. persica* in methanol extracts of leaves was 252.770 mg/g, whereas the total flavonoid contents were estimated to be 57.94 mg/g [19], which is higher than our findings. Variation in our TFC and TPC results compared with the findings of the majority of previous studies might have been due to differences in the agro-climatic, geographical, and seasonal conditions.

### 3.4. DPPH Free Radical Scavenging Assay

The free radical scavenging activity of various extracts (10 µg/mL) was measured by the DPPH radical scavenging assay, and the results are presented in Table 1. The fruit extract of *S. persica* showed maximum radical scavenging activity (54.3%), whereas the aerial parts of *S. oleoides* showed minimum radical scavenging activity (46.9%) when compared with the synthetic antioxidant BHT (60.8%). Statistical analysis showed the significant (*p* ≤ 0.05) differences in the radical scavenging potential of *S. persica* fruit extracts from other extracts.

The DPPH free radical scavenging capacity increases when extract concentration increases due to increases in the concentration of phenolic compounds [5]. Saleem et al. and Noumi et al. [23,24] reported the strongest scavenging DPPH assay results (51.66%) of methanolic extracts of *S. oleoides* aerial parts, which were comparable to our results. Kumari et al. [25] reported the IC_50_ for DPPH of the *S. persica* fruit (IC_50_ 307.06 µg crude methanol extract). Souri et al. [26] investigated the antioxidant activity and free radical scavenging activity on DPPH of 13 medicinal plants traditionally used in Iran. They reported that methanolic extracts of *S. persica* exhibit free radical DPPH scavenging activity with an IC_50_ value of 37.19, which is contrary to our results. The variation in DPPH radical scavenging activity might be due to different plant species, geographic regions, as well as different months for sample collection.

### 3.5. In Vivo Study on Animal Model

#### 3.5.1. Anti-Inflammatory Activity

The anti-inflammatory activities of aerial part and fruit extracts of *S. persica* and *S. oleoides* were evaluated, and the results are presented in Table 3. At zero hour, mean inflammation (mm ± SD) was non-significant (*p* > 0.05) in all groups before the start of treatment.

The inflammation was significantly reduced (*p* ≤ 0.05) in all treated groups, G1, G2, G3, and G4, and the positive control (PC) as compared with the negative control (NC) group one hour after the oral administration of methanolic extracts. However, reductions in inflammation were non-significant (*p* > 0.05) among all treated groups one hour after the oral administration of extracts. The inflammation was significantly reduced (*p* ≤ 0.05) in all treated groups, G1, G2, G3, and G4, and the positive control group (PC) as compared with the negative control (NC) groups 2 h after the oral administration of methanol extract. However, the anti-inflammatory response was significantly reduced in G4 as compared with NC and G1 2 h after the oral administration. Inflammation was also significantly reduced in G1, G2, and G3 as compared with NC, but was non-significant between groups. The inflammation was significantly reduced (*p* < 0.05) in all treated groups, G1, G2, G3, and G4, and PC, as compared with the NC group 3 h after the oral administration of extracts. Inflammation was significantly reduced in G4 as compared with G1, G2, and G3, but was comparable to PC. Inflammation was also significantly reduced in G1, G2, and G3 as compared with NC, but less than G4 and PC. The inflammation was significantly reduced (*p* < 0.05) in all treated groups, G1, G2, G3, and G4, and PC, as compared with NC four hours after the oral administration of methanol extract. Inflammation was significantly reduced (*p* < 0.05) in G4 as compared with NC, G2, and G3. It could be concluded that the fruits of *S. persica* are more potent than the other extracts used in the study.

Polyphenols are secondary metabolites that have antioxidant capacities in addition to antiallergenic, anticancer, anti-inflammatory, anti-thrombotic, and anti-mutagenic properties [27]. The results obtained by BenSaad et al. [28] clearly indicate that ellagic acid, gallic acid, and punicalagin A & B isolated from *P. granatum* inhibited the production of NO, PGE2, and IL-6 in LPS-induced RAW267.4 macrophages. In the present study, gallic acid may have been the compounds responsible for the remarkable anti-inflammatory effect showed by *S. presica* fruit. Catechins can also inhibit the infiltration and proliferation of immune-related cells and regulate inflammation and oxidative reactions by interaction with multiple inflammation-related and oxidative-stress-related pathways [29]. Catechin was the major compound found in *S. persica* fruit and detected in all extracts by Rp-HPLC, which is why it could also be the reason for the anti-inflammatory effect shown by plant extracts. Ibrahim et al. [30] investigated the anti-inflammatory effect of aqueous alcoholic crude extract and the ethyl acetate extract of miswak sticks (*S. persica*) in carrageenan-induced rat paw edema. The inhibition percentage of inflammation was 17% for crude extract and 27% for ethyl acetate extract. Natubhai et al. [21] studied the anti-inflammatory effect of *S. oleoides* leaf extract. The alcoholic extracts of *S. oleoides* at a dose of 200 and 400 mg/kg, reduced the paw edema induced by carrageenan by 46.70 and 81.70%, respectively, whereas the water extract reduced the paw edema by 51.16 and 83.30, respectively. No reports are available in the literature on the anti-inflammatory activity of aerial parts and fruits of *S. persica* and *S. oleoides*. Baba et al. [31] investigated the anti-inflammatory activity of ethanol, acetone, and water extract of leaves, stem bark, and fruit peels of *S. persica*. The anti-inflammatory activity was compared with the standard drug (Diclofenac) at a dose of 10 mg/kg. Diclofenac showed values of 87.71 and 82.85 at a dose of 300 mg/kg, respectively.

#### 3.5.2. Analgesic Activity

The analgesic activity of aerial part and fruit extracts of *S. oleoides* and *S. persica* was investigated using a mouse model, and the results are presented in Table 4. Mice treated with normal saline (control) did not show any significant difference in the reaction time on the experiment throughout the 240 min observation. The longest reaction time for the treated groups with the hot plate method was 180 min. The analgesic effects of Diclofenac sodium and different plant extracts could be seen from the maximum possible analgesia (MPA) graph in Figure 2. The MPA remained elevated during the observation period, reaching its peck at 180 min. The MPA of fruit extracts of *S. persica* was 53.44% that is better than PC group (52.98%). Aerial part extracts of both *S. oleoides* and *S. persica* showed significantly (*p* ≤ 0.05) less MPA. Overall, the fruit extract of *S. persica* produced an excellent analgesic activity at 180 min.

Hooda and Pal [32] reported the analgesic activity of hydroalcoholic extract of *S. persica* root extract on a mouse and rat model. The analgesic activity of albino mice and albino rats was evaluated using Eddy’s hot plate method. In Eddy’s hot plate method, the highest analgesic activity was observed at an oral concentration of 400 mg/kg for 90 min. Our results were correlated with those of Hoor et al. [33], who suggested that extracts of *S. persica* possess analgesic activity. The crude extract was used in three doses of 300 mg/kg, 500 mg/kg, and 700 mg/kg of tested animals against standard aspirin, and found to be maximum at 120–150 min.

## 4. Conclusions

The results obtained in the present study showed the comparative assessment of TPC and TFC, and antioxidant, anti-inflammatory, and analgesic activities of *S. persica* and *S. oleoides* aerial parts and fruit extracts. *S. persica* fruit extract showed the maximum antioxidant activities in terms of TPC, TFC, and DPPH radical scavenging. Based on the results, it is concluded that fruit extracts of *S. persica* have a major role in the reduction in inflammation and exhibit better analgesic activity as well. This was due to the high antioxidant results of *S. persica* fruit extract as compared with extracts of other plant parts that were investigated, and also due to the high presence of gallic acid and catechin, as shown by the quantification of phenolic acids and flavonoids, performed using Rp-HPLC. The present study also offers the scientific evidence to use fruits of *S. persica* in treating inflammation and provides a framework for the use of *S. persica* fruits as natural antioxidants after further clinical trials. There is still a dearth of evidence for exploring its importance. However, further investigations are ongoing to determine the exact phytoconstituents that are responsible for the biological activities of methanol extracts of Salvadora species.

## Figures and Tables

**Figure 1 life-12-01446-f001:**
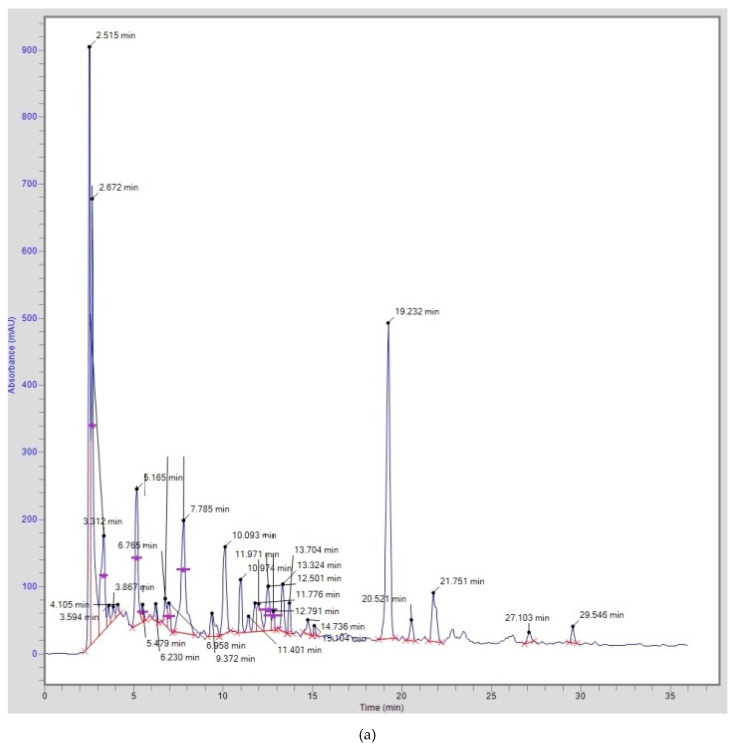
Typical HPLC chromatograms showing the separation of phenolic acid and flavonoids from (**a**) *S. oleoides* aerial parts; (**b**) *S. oleoides* fruit; (**c**) *S. persica* aerial parts; and (**d**) *S. persica* fruit extract.

**Figure 2 life-12-01446-f002:**
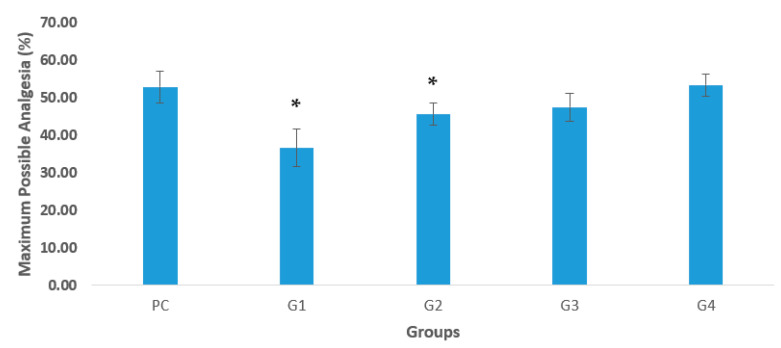
Maximum possible analgesia (MPA) (%) representing the effect of the drugs administered into mice, evaluated by the hot plate method. NC, normal control group; PC; positive control (Diclofenac sodium 10 mg/kg body weight); and treated groups: G1 (aerial part extract of *S. oleoides* 250 mg/kg body weight), G2 (aerial part extract of *S. persica* 250 mg/kg body weight), G3 (Fruit of *S. oleoides* 250 mg/kg body weight), and G4 (fruit of *S. persica*, 250 mg/kg body weight). * indicates significant (*p* ≤ 0.05) differences from the PC group.

**Table 1 life-12-01446-t001:** Yield, TPC, TFC, and DPPH radical scavenging capacity of *S. oleoides* and *S. persica* aerial parts and fruit extracts.

Assays	*S. oleoides*		*S. persica*		BHT
	Aerial Parts	Fruits	Aerial Parts	Fruits	
Yield (g/100 g of dry weight)	8.15 ± 0.48 ^a^	14.90 ± 0.74 ^b^	19.60 ± 0.98 ^c^	16.20 ± 0.81 ^b^	---
TPC (mg/g, measures as gallic acid equivalent)	11.90 ± 0.50 ^a^	19.20 ± 0.96 ^c^	16.20 ± 0.81 ^b^	22.20 ± 1.11 ^d^	---
TFC (mg/g, measured as catechin equivalent)	1.72 ± 0.09 ^a^	5.54 ± 0.28 ^c^	2.44 ± 0.12 ^b^	6.17 ± 0.31 ^c^	---
DPPH scavenging (%) by 10 µg/mL extract solution	46.90 ± 2.34 ^a^	52.70 ± 2.63 ^b^	51.60 ± 2.58 ^ab^	54.30 ± 2.72 ^b^	60.80 ± 3.04 ^e^

The values are the mean ± SD of three independent experiments. Different alphabet letters in superscript show significant (*p* ≤ 0.05) differences among different extracts.

**Table 2 life-12-01446-t002:** Composition (mg/100 g) of phenolic acids and flavonoids from *S. oleoides* and *S. persica* aerial parts and fruit extracts, obtained using Rp-HPLC.

Compounds	S. oleoides		S. persica	
	Aerial Parts	Fruits	Aerial Parts	Fruits
Gallic acid	254.4 ± 11.1 ^a^	1265.0 ± 36.0 ^d^	580.4 ± 20.3 ^b^	942.4 ± 24.8 ^c^
Hydroxy benzoic acid	254.8 ± 12.3 ^a^	727.0 ± 21.2 ^c^	388.9 ± 9.7 ^b^	732.9 ± 16.5 ^c^
Chlorogenic acid	526.4 ± 17.3 ^a^	1786.0 ± 44.0 ^c^	540.7 ± 21.9 ^a^	1473.0 ± 48.0 ^b^
Caffeic acid	16.4 ± 0.8 ^a^	14.6 ± 0.8 ^a^	165.9 ± 5.4 ^c^	54.0 ± 1.3 ^b^
Syringic acid	23.0 ± 1.2 ^b^	71.6 ± 3.0 ^c^	---	15.4 ± 0.8 ^a^
Vanillic acid	183.1 ± 4.1 ^b^	176.8 ± 5.3 ^b^	---	86.5 ± 4.1 ^a^
*p*-Coumeric acid	136.4 ± 5.8 ^ab^	131.9 ± 5.2 ^a^	257.4 ± 10.3 ^c^	145.7 ± 5.9 ^b^
Salicylic acid	239.3 ± 10.1 ^c^	151.8 ± 5.9 ^b^	167.5 ± 6.5 ^b^	119.8 ± 5.2 ^a^
Sinapic acid	69.3 ± 3.1 ^c^	13.3 ± 0.8 ^a^	88.3 ± 3.2 ^d^	45.9 ± 2.5 ^b^
Ferulic acid	22.0 ± 1.4 ^a^	27.9 ± 0.9 ^b^	21.6 ± 1.0 ^a^	52.2 ± 2.3 ^c^
Ellagic acid	22.0 ± 1.4 ^b^	26.5 ± 1.3 ^c^	52.5 ± 2.0 ^d^	17.5 ± 0.8 ^a^
Cinamic acid	390.6 ± 12.1 ^b^	---	471.6 ± 14.3 ^c^	33.0 ± 1.4 ^a^
Benzoic acid	20.9 ± 1.2 ^b^	---	24.5 ± 1.1 ^c^	9.5 ± 0.6 ^a^
Catechin	131.5 ± 5.90 ^c^	72.0 ± 2.7 ^a^	578.5 ± 25.2 ^d^	117.2 ± 4.9 ^b^
Rutin	302.7 ± 10.23 ^d^	59.0 ± 2.2 ^a^	254.3 ± 10.0 ^b^	275.1 ± 9.17 ^c^
Myricetin	332.0 ± 12.50 ^b^	28.0 ± 1.0 ^a^	410.5 ± 17.8 ^c^	331.8 ± 17.6 ^b^
Quercetin	24.16 ± 0.89 ^b^	---	21.7 ± 0.9 ^a^	21.0 ± 0.9 ^a^
Kaempferol	---	---	26.2 ± 1.3 ^a^	24.7 ± 1.2 ^a^

The values are the mean ± SD of three independent experiments. Different alphabet letters in superscript show significant (*p* ≤ 0.05) differences.

**Table 3 life-12-01446-t003:** Rat paw diameters and edema inhibition percentages (%) of different treatment groups at different time intervals.

Groups	Paw Diameter (mm) (Edema Inhibition %)				
	Initial Value	1 h	2 h	3 h	4 h
NC	4.14 ± 0.06 ^a^	8.24 ± 0.21 ^c^	6.18 ± 0.29 ^c^	5.68 ± 0.26 ^e^	5.21 ± 0.27 ^e^
PC	4.17 ± 0.07 ^a^ (0.92%)	6.11 ± 0.24 ^a^ (25.73%)	5.21 ± 0.28 ^b^ (15.60%)	4.24 ± 0.10 ^b^ (25.35%)	4.06 ± 0.03 ^b^ (22.07%)
G1	4.14 ± 0.07 ^a^ (0.54%)	6.68 ± 0.19 ^ab^ (19.05%)	5.54 ± 0.13 ^ab^ (10.36%)	5.23 ± 0.04 ^d^ (7.92%)	4.15 ± 0.14 ^bc^ (20.15%)
G2	4.17 ± 0.01 ^a^ (0.82%)	6.31 ± 0.05 ^ab^ (23.42%)	5.17 ± 0.24 ^b^ (16.34%)	4.70 ± 0.14 ^c^ (17.25%)	4.55 ± 0.16 ^d^ (12.67%)
G3	4.27 ± 0.07 ^a^ (0.38%)	7.08 ± 0.38 ^b^ (14.08%)	5.24 ± 0.07 ^b^ (15.21%)	4.80 ± 0.15 ^c^ (15.49%)	4.35 ± 0.27 ^cd^ (16.51%)
G4	4.12 ± 0.04 ^a^ (0.48%)	6.44 ± 0.27 ^a^ (21.84%)	4.61 ± 0.22 ^a^ (25.40%)	4.16 ± 0.08 ^a^ (26.76%)	3.69 ± 0.15 ^a^ (29.17%)

The values are reported as the mean ± SD. Different alphabet letters in superscript show significant (*p* < 0.05) differences among different rat groups. NC, negative control (0.1 mL albumin); PC, positive control (Diclofenac sodium 10 mg/kg body weight) and treated groups: G1 (aerial part extract of *S. oleoides,* 250 mg/kg body weight), G2 (aerial part extract of *S. persica* 250 mg/kg body weight), G3 (Fruit extract of *S. oleoides* 250 mg/kg body weight), and G4 (fruit extract of *S. persica* 250 mg/kg body weight).

**Table 4 life-12-01446-t004:** Analgesic activity of the test drug in hot plate test.

Groups	Reaction Time (Minutes)				
	Start Time	60	120	180	240
NC	3.41 ± 0.21 ^a^	3.44 ± 0.32 ^a^	3.46 ± 0.32 ^a^	3.45 ± 0.32 ^a^	3.45 ± 0.33 ^a^
PC	3.43 ± 0.29 ^a^	5.53 ± 0.43 ^b^	6.89 ± 0.62 ^c^	6.92 ± 0.65 ^bc^	5.42 ± 0.49 ^b^
G1	3.45 ± 0.37 ^a^	5.61 ± 0.54 ^b^	5.52 ± 0.47 ^b^	5.85 ± 0.53 ^b^	5.43 ± 0.47 ^b^
G2	3.46 ± 0.31 ^a^	6.21 ± 0.59 ^b^	6.11 ± 0.57 ^bc^	6.44 ± 0.58 ^bc^	5.98 ± 0.54 ^b^
G3	3.47 ± 0.34 ^a^	6.33 ± 0.58 ^b^	6.23 ± 0.58 ^bc^	6.56 ± 0.59 ^bc^	6.12 ± 0.56 ^b^
G4	3.51 ± 0.32 ^a^	6.45 ± 0.58 ^b^	6.62 ±0.59 ^b^	6.95 ± 0.64 ^c^	6.31 ± 0.59 ^b^

The values are presented as the mean ± SD. Different alphabet letters in superscript show significant (*p* ≤ 0.05) differences among normal and treatment groups. NC, negative control (0.1 mL albumin); PC, positive control (Diclofenac sodium 10 mg/kg body weight); and treated groups: G1 (aerial part extract of *S. oleoides,* 250 mg/kg body weight), G2 (aerial part extract of *S. persica* 250 mg/kg body weight), G3 (Fruit extract of *S. oleoides* 250 mg/kg body weight), and G4 (fruit extract of *S. persica* 250 mg/kg body weight).

## Data Availability

Not applicable.

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
