# Peer review of "Variation in Phenolic Profile, Antioxidant, and Anti-Inflammatory Activities of *Salvadora oleoides* Decene. and *Salvadora persica* L. Fruits and Aerial Part Extracts"

_life, 2022, doi:10.3390/life12091446_

Round 1

Reviewer 1 Report

Dear Authors,

Despite all advantages of the research, the manuscript containes some shortcomings; and needs revision.

Table 1, data have to be presented  not only by units, but also by decimal values (with one or two numbers after dot).

Table 2, the expression of all data must to be of the same mode (two or one number after dot).

Results' section, the first sentence needs correction of style: too much of "acid" words.

Author Response

All these suggestions from reviewer 1 are welcome and all suggested changes have been answered and an itemized list of responses are attached. We thank reviewer 1 for improving this article with his valuable suggestions.

Reviewer 2 Report

 Salvadora persica L. and Salvadora oleoides: writ the names in italic.

Hplc chromatogram of Salvadora persica??

plant collection year?

amount of plants?

What are the meaning of a, b, c ,d superscripts in each table, kindly explain it

Author Response

All responses are address in itemized rebuttal letter and we appreciate all suggestions made by reviewer 2.

Round 2

Reviewer 2 Report

Accept